# The Social Life of Pigs: Changes in Affiliative and Agonistic Behaviors following Mixing

**DOI:** 10.3390/ani12020206

**Published:** 2022-01-16

**Authors:** Carly I. O’Malley, Juan P. Steibel, Ronald O. Bates, Catherine W. Ernst, Janice M. Siegford

**Affiliations:** 1Department of Animal Science, Michigan State University, East Lansing, MI 48824, USA; steibelj@msu.edu (J.P.S.); batesr@msu.edu (R.O.B.); ernstc@msu.edu (C.W.E.); siegford@msu.edu (J.M.S.); 2Department of Fisheries and Wildlife, Michigan State University, East Lansing, MI 48824, USA

**Keywords:** aggression, affiliation, play, nosing, non-agonistic contact

## Abstract

**Simple Summary:**

Aggression in pigs is a major welfare concern in the pig industry as pigs fight when mixed into new social groups. Most attempts to solve this problem have focused on reducing agonistic behavior. However, another approach could be to study positive social behaviors in pigs and examine how these relate to aggressive behaviors. Understanding the full social experience of pigs and how affiliative behaviors may mitigate aggression could lead to better selection and management of pigs. The purpose of this study was to gain insight into the prevalence and change in performance of potentially affiliative behaviors in pigs after a mixing event, as well as how these behaviors relate to the amount of aggression shown. In this study, the prevalence of suspected affiliative behaviors changed for up to 9 weeks following mixing pigs into a new social group, with nosing decreasing following mixing, and play and non-agonistic contact increasing. All affiliative behaviors were negatively related to agonistic behavior at mixing but nosing and play behaviors were positively related to agonistic behavior in the weeks after mixing. Non-agonistic physical contact was consistently related to less agonistic behavior and therefore could be an indicator of positive social relationships between pigs. Further research could explore how to promote non-agonistic contact and other positive social behaviors among pigs to help reduce agonistic behaviors.

**Abstract:**

This study investigated potentially affiliative behaviors in grow-finish pigs, how these behaviors changed over time and their relationship to agonistic behaviors. A total of 257 Yorkshire barrows were observed for agonistic (reciprocal fights, attacks) and affiliative (nosing, play, non-agonistic contact) behaviors after mixing (at 10 weeks of age), and weeks 3, 6, and 9 after mix. The least square means of affiliative behaviors were compared across time points. Relationships among affiliative and agonistic behaviors were assessed using generalized linear mixed models. Non-agonistic contact with conspecifics increased until week 6 then remained stable between weeks 6 and 9. Nosing was highest at mix, then decreased in the following weeks. Play was lowest at mix and highest at week 3. Affiliative behaviors were negatively related with aggression at mix (*p* < 0.001). Pigs who engaged in play and nosing behaviors were more likely to be involved in agonistic interactions in the weeks after mixing (*p* < 0.05), while pigs engaging in non-agonistic contact were less likely to be involved in agonistic interactions (*p* < 0.001). There appear to be relationships between affiliative and agonistic behaviors in pigs, with contact being the most predictive of less aggression. Future studies could focus on promoting positive non-agonistic contact in unfamiliar pigs as a way to mitigate aggressive interactions.

## 1. Introduction

Group housing is common in the commercial pig industry for non-breeding animals and provides a number of benefits to pigs such as increased space allowance per pig, the ability to perform more natural behaviors, and interaction with conspecifics. However, group housing also presents major welfare concerns, most notably increased aggression for up to 48 hours after pigs are mixed to form uniform groups based on sex and weight [1,2] and potentially chronic levels of aggression if stable social groups are not established [3]. Chronic social stress can have prolonged negative effects on pig welfare, including disruptions to growth and immune function [3]. Management interventions available to producers to address this issue do not successfully mitigate aggression altogether, but rather reduce severity and duration of high intensity aggression or delay its onset [4,5]. Research has focused on finding a genetic component of social aggression to allow producers to breed pigs more suitable for group housing systems [1,6,7,8]. However, little research has examined what successful group housing looks like, particularly in regard to affiliative behaviors occurring among pigs that could be indicators of positive social relationships. In a study comparing affiliative behaviors among pigs kept with littermates and those mixed with unfamiliar pigs, there were no differences in prevalence of affiliative behaviors for up to 7 weeks after weaning, although pigs kept with littermates had higher growth rates than mixed pigs [9]. Commercially raised pigs typically do not receive early life socialization needed to develop social skills, which helps reduce aggression later on [10]. Social genetic effects indicate a link between social behaviors and growth rate [11], therefore, having a better understanding of how pigs display affiliative behavior in stable social groups could allow breeding programs to reduce social aggression by breeding pigs that are more positively social, and able to read and respond to social cues, rather than using indicators of negative social interactions and aggression, such as skin lesions [1,6,7,8]. 

Studies of pig social behavior have primarily focused on aggression, but pigs, like all gregarious species, have a wide range of behaviors that are meant to promote strong social bonds [12]. Non-agonistic physical contact, gentle nosing, and play have previously been studied as possible affiliative behaviors in pig social groups [3,9,13,14,15]. However, evidence on which behaviors are truly affiliative, in that they will promote positive social interactions and stable social groups, have not been well studied in domestic animals, including pigs [13,14]. For example, nosing is often considered to be an affiliative behavior in pigs, similar to social grooming and gentle touch in other species [16]. Pigs that receive nosing had improved growth rate compared to pigs that received oral manipulation, like tail or ear biting, suggesting a relationship to dominance [11]. However, a follow-up study found no relationship between nosing and dominance relationships, or harmful behaviors, making the function of nosing as a social behavior ambiguous [16]. Understanding the full range of behaviors pigs exhibit as they integrate into new social groups is important for properly managing pigs and mitigating aggression. 

The objectives of this research were to quantify potentially affiliative behaviors in grow-finish pigs including play, nosing, and contact to conspecifics at four time points after mixing (at mix, 3, 6, 9 weeks after mix), and determine levels of affiliation and aggression in recently mixed and stable social groups. At 6 weeks after mix, pigs were moved to a new pen, keeping the pen group intact, to investigate the effects of a minor change in environment on affiliative and agonistic behaviors. The hypotheses were that pigs would display more affiliative behavior in stable social groups, particularly when introduced to a new pen, and that displaying more affiliative behaviors would be related to less agonistic behavior.

## 2. Materials and Methods

### 2.1. Animals and Housing

All animals included in this study were housed at the Michigan State University Swine Teaching and Research Center in East Lansing, MI, USA. A total of 257 purebred Yorkshire barrows (castrated males) housed across 20 pens were observed starting at 10 weeks of age (approximately 23 kg) when they were mixed into new groups in finisher pens (4.83 m × 2.44 m). A commercially formulated diet specific to the nutritional requirements of pigs at that production stage was provided ad libitum [17] from self-feeders with no more than 10 pigs per space. Pigs also had ad libitum access to water from nipple in cup drinkers, with one drinker available in each pen. The pigs received full incandescent light for 8 hours per day, and half-light from auxiliary incandescent bulbs for 16 hours per day. 

To create the new groups, pigs were mixed into same sex groups with pigs of similar weight. The new social groups consisted of 3–5 groups of pigs from different nursery pens for a total of 10–15 pigs per finisher pen. Thus, each pig was grouped with 2–5 previous penmates from their nursery pen and 10–13 pigs that were from other nursery pens. To test the effects of a novel environment (i.e., a stressor) on affiliative behavior in a stable group of pigs, pigs were moved to a similar but unfamiliar pen 6 weeks after mix. Pigs remained in this pen until the end of this study. The new pens were in the same room and had the same resources, design, and dimensions as their original pen, and pigs were housed with the same social group. The pens that groups of pigs were moved into were randomly selected.

### 2.2. Behavioral Observations

Behavioral observations were conducted immediately after mixing at 10 weeks of age, then at 3, 6, and 9 weeks after mixing when groups are typically considered relatively stable. Observations were made using video recorded by a ceiling-mounted camera (Clinton Electronics VF540 Bullet Cameras; Loves Park, IL, USA) above each pen that was connected to a digital video recorder (Geovision 1480A; Taipei, Taiwan). 

For identification purposes, the back of each pig was marked with a unique number using a non-toxic permanent marker. Pigs were observed for 4 consecutive hours in the afternoon at each of the 4 time points (immediately after being mixed, 3, 6 [after pigs were moved into the new pen], and 9 weeks (wk) after mix) to capture affiliative and aggressive behaviors. 

Affiliative behaviors included play, nosing, non-agonistic physical contact, and aggression. Duration (seconds (s)) of play behaviors were recorded using all-occurrence sampling [18] included scamper, pivot, head toss, flop, and paw from the ethogram reported in [19]. Play was scored by 2 trained observers (≥80% inter- and intra-observer reliability). Social play was not included due to inability to distinguish social play from aggression. Any behaviors displayed during bouts of play that could be defined as aggressive (i.e., head knocks towards penmates) was scored as such. Nosing was defined as any interaction where a pig touched its nose to a conspecific and duration (s) was recorded using all-occurrence sampling [18]. The type of nosing and the initiator of nosing were recorded. Type of nosing included nose-head, nose-body, and nose-nose. Nose-head (NH) was defined as the nose of the initiator pig touching the neck, head, or ears of the receiving pig. Nose-body (NB) was defined as the nose of the initiator pig touching anywhere posterior to the base of the neck, including the back, rear, legs, and tail. Nose-nose (NN) was defined as pigs mutually touching their noses. Initiator and receiver were recorded for NH and NB behaviors, but NN was considered a mutual behavior due to difficulties in identifying the initiator, therefore initiator and receiver were not recorded. Nosing was scored by 1 trained observer. Pigs were also observed for non-agonistic physical contact with conspecifics using scan-sampling [18] every 10 minutes for the same 4-hour time period used for the other behaviors. The contact variable was scored as the number of unique pigs the focal pig was in physical contact with without overt aggression occurring at each scan interval divided by the total number of scan intervals (25). Physical contact referred to any body contact between animals. Non-agonistic contact was scored by 3 trained observers (≥80% interobserver reliability). 

Duration (s) of aggressive behaviors was recorded using all-occurrence sampling [18] and included reciprocal aggression (fights), non-reciprocal aggression (rest during fights, withdrawals, attacks), head knocks, single bites, and presses (ethogram provided in Table 1). Initiator and receiver of the aggression were recorded for one-sided behaviors such as attacks and head knocks. Data were summarized into duration of total aggression (defined as any aggressive interaction regardless of whether the pig with the initiator or receiver), and duration of initiated aggression (defined as aggressive interactions where the pig was the initiator, used as a measure for aggressiveness) for each pig. Aggressive behaviors were scored by 21 trained observers (≥80% inter- and intra-observer reliability). 

### 2.3. Statistical Analyses

Data analyses were completed using R [20].

The observational unit of the analysis was individual pig within pen. A linear mixed effect model was fit with period as a fixed effect and pen as a random effect. Affiliative behaviors were compared across time points using least square means. Tukey’s honest significance test was used to obtain adjusted *p*-values. All variables were assessed for normality by visual inspection of quantile-quantile plots and using the Shapiro–Wilk test. Duration (s) of nose-head, nose-body, nose-nose, total nosing, total aggression, total initiated aggression, and play were transformed for normality using a log_10_ + 1 transformation. 

Affiliative behaviors were compared to agonistic behaviors using generalized linear mixed models fitted for each agonistic measure (duration of total aggression and initiated aggression) and each time period (mix and 3, 6, 9 weeks after mix). Models were also fitted to test the effects of affiliative behaviors performed at mix with aggression occurring at each of the later time points. The models included the affiliative behaviors of play, nosing (scaled prior to analysis due to differences in the scales of the measurements for nosing, play, and non-agonistic contact, and due to the vast differences between duration of nosing and play), and non-agonistic contact as fixed effects, and pen as random effect. 

## 3. Results

### 3.1. Quantifying Affiliative Behaviors across Time and When Moved to a New Pen

#### 3.1.1. Aggression

The least square means of duration of aggression (s), including total aggression and initiated aggression, are presented in Figure 1. The most total and initiated aggression occurred at mix. Total aggression decreased from mix in week (wk) 3, wk 6, and wk 9. The amount of initiated aggression was similar between wk 3 and wk 6 but decreased in wk 9. There appeared to be no effect of moving to a new pen at wk 6 on amount of time spent in aggressive behaviors.

#### 3.1.2. Nosing

The least square means of duration of nosing (s), including nose-body, nose-head, nose-nose, and total nosing across time points are presented in Figure 2. Overall, pigs engaged in nosing more immediately after mixing, and this behavior decreased throughout the remainder of the study period. Nosing behavior directed towards the body and head of another pig was performed more than nose-nose behavior. Pigs did not spend more or less time nosing after being moved to a new pen at wk 6, in comparison to the time spent nosing observed at wk 3 and wk 9.

#### 3.1.3. Non-Agonistic Physical Contact

The least square means of non-agonistic physical contact across time is presented in Figure 3. The time pigs spent in contact with other pigs was lowest immediately after mix, highest in wk 3 and 6, then decreased in wk 9. Moving to a new pen at wk 6 did not appear to cause a change in non-agonistic physical contact among pigs.

#### 3.1.4. Play

The least square means of duration of play behavior are presented in Figure 4. Play behavior was lowest after mixing into new social groups, then peaked at wk 3. Play behavior then occurred for less time in wk 6 and wk 9 but was consistent between the two time periods suggesting there was no effect of moving to a new pen on play behavior. 

### 3.2. Comparisons between Affiliative and Agonistic Behaviors

The relationships between affiliative and agonistic behaviors within each time point (mix, wk 3, wk 6, and wk 9) are presented in Table 1. At mixing, pigs that spent more time engaged in nosing, non-agonistic contact, and play spent less time involved in agonistic behaviors (total aggression and total initiated aggression). Pigs that spent more time in non-agonistic contact spent less time involved in agonistic interactions at all time points. At weeks 3, 6, and 9, pigs that spent more time engaged in nosing behaviors spent more time involved in agonistic interactions. This was true for all nosing behaviors, including nose-head, nose-body, and nose-nose, therefore only total nosing is presented in Table 2 Pigs that spent more time playing also spent more time involved in agonistic interactions until wk 6. There was no relationship between time spent in play and in agonistic behaviors at wk 9. 

Predictive relationships between amount of time spent on affiliative behaviors immediately after mixing and amount of time engaged in aggression at wk 3, wk 6, and wk 9 after mixing were also investigated, and there were no relationships between duration of affiliative behaviors at mix and duration of agonistic behaviors at later time points (*p* > 0.115).

## 4. Discussion

Efforts to address the welfare concerns present in pigs due to social aggression have mainly focused on reducing agonistic interactions among pigs. However, solely focusing on reduced aggression may not be an effective approach for two reasons. First, less aggression among pigs upon being grouped with unfamiliar animals has been shown to lead to chronic aggression and concomitant increases in physiological stress responses and decreases in immune function [3,21]. Second, animals that have evolved to live in groups, including pigs, display a wide range of social behaviors, both affiliative and agonistic, to promote social bonds that strengthen group cohesion and stability [12]. Having a better understanding of these social behaviors in pigs may help producers and researchers identify and implement behavioral management techniques that not only reduce agonistic behaviors, but also promote affiliative behaviors and positive welfare. The objectives of this study were to quantify potentially affiliative behaviors in grow-finish pigs at four time points following a mixing event and to compare these affiliative behaviors to levels of agonistic behaviors. The potentially affiliative behaviors investigated in this study were non-agonistic physical contact, nosing, and play. We also aimed to investigate the effects of a minor stressor, moving pigs to a new pen, on affiliative and agonistic behaviors. Our hypotheses were that pigs would display more affiliative behavior in stable social groups, particularly when introduced to a new pen, and that more time spent in affiliative behaviors would correspond with less agonistic behavior. 

The time pigs spent in non-agonistic contact with conspecifics increased after mix, but decreased again at week 9. Non-agonistic body contact or proximity to conspecifics has been used as a measure of affiliation in social animals, as animals tend to stay near or touch familiar or preferred conspecifics [14,15]. Pigs can take weeks to settle into a new environment, and elevated levels of aggression can persist in a group for at least 3 weeks following a mixing event [3]. It is likely that the increase in time spent in affiliative contact between week 3 and week 6 was due to pigs establishing social groups. Immediately after a mixing event, the pigs in this study were also found to remain in affiliative physical contact with familiar pigs over unfamiliar pigs, demonstrating that they preferentially associate with some pigs more than others in positive ways [22]. On the day of mixing, 53% of the dyads of familiar pigs showed preferential associations, while only 9% of unfamiliar pigs showed preferential associations. However, 3 days after mixing, preferential associations between familiar and unfamiliar pigs were similar at 20% and 18%, respectively [22]. In the results presented here, pigs that engaged in more non-agonistic contact spent less time involved in agonistic interactions at all time points, suggesting that physical contact may be a valid measure of affiliation in pigs. Camerlink et al. [14] did not find a relationship between aggression and social proximity of pigs. However, they did not parse out differences between total or initiated aggressive interactions and looked specifically at distances between pigs, not proportion of time pigs spent in non-agonistic physical contact with each other. A limitation of using non-agonistic contact or proximity as a measure of affiliation is that the pigs are housed in a restricted space and as pigs grow, they have less opportunities to spatially disperse. 

Pigs nosed at all time points but the amount of time observed nosing decreased from the time of mixing over the course of the 9-week study period. The motivation behind nosing between pigs is not well understood. Camerlink et al. [11] found a positive link between growth rate and pigs that receive nosing, suggesting a potential link between nosing and social dominance. However, these results were not supported by Camerlink et al. [16] where no clear benefits or motivation was found for giving or receiving nosing behavior relative to dominance relationships. In the present study, most interactions between pigs were preceded by or followed by nosing. Nosing is a way for pigs to detect cues from their environment and is used in social recognition and communication [16,23]. There are instances where nosing in pigs is considered harmful, for example when it leads to belly nosing, and tail or ear biting [11]. In the present study, distinguishing between affiliative nosing and harmful nosing was not always possible. For example, some incidences of nosing were of long duration and directed at the belly or ear, especially in the stable time points, and could have been a form of stereotypic or harmful oro-nasal behavior [11]. If nosing is an affiliative behavior used for social recognition and communication as suggested by Camerlink et al. [11], then nosing would be predicted to increase at week 6 as pigs adjust to a novel pen. This did not appear to be the case in our study. At 3, 6, and 9 weeks after mixing, nosing was positively related to duration of total aggression and duration of initiated aggression, therefore, it is possible that nosing was a form of stereotypic or displaced exploratory behavior due to the lack of stimuli in the environment [11]. Thus, the social function of nosing in pigs remains ambiguous with no clear role in promoting positive social interactions and stable social relationships. Therefore, at present, nosing is not a useful measure of social relationships among pigs.

The time pigs spent playing was lowest at mix and peaked at week 3. Play is often seen in juvenile animals and is thought to be important in the development of behavioral and physical skills needed as an adult [24]. Play is often proposed as a measure of positive affective states in animals, with stressful situations typically (but not always) causing a decrease in play [25]. Mixing is stressful for pigs, so the low amount of time spent playing at that time point is in line with play as an indicator of positive welfare. Play in pigs peaks between 2 to 6 weeks of age [23,26]. The pigs in this study were 10 to 19 weeks of age, which may explain the low occurrence of play overall, and the further decrease in play in weeks 6 and 9. However, introducing pigs to novel or bigger environments elicits play behavior in pigs older than 6 weeks, including in mature sows [23]. Yet, in the present study, an increase in play was not seen when pigs were moved to a novel pen at week 6. The barren environment the animals were housed in might also have contributed to the low occurrence of play recorded in this study [23]. In this study, we also only recorded locomotor play, as social play can be difficult to distinguish from aggression on video recordings. In weeks 3 and 6, play was associated with duration of total aggression and at week 3, also with duration of initiated aggression but due to the low amount of play observed, the relationships between play and aggression should be interpreted with caution. While it is generally assumed that play occurs in the absence of stress [23], evidence suggests that play can also act as a coping mechanism for individuals in stressful conditions [25]. The presence of play immediately after mixing may be a coping mechanism for pigs avoiding agonistic interactions, thus causing them to be the target of aggressive interactions at weeks 3 and 6 after mixing [21,27], but this connection needs to be investigated further. Another explanation for the positive relationship between play and aggression at weeks 3 and 6 is that inactivity was the most common behavior seen in the observed pigs, with inactivity increasing throughout the study period [28]. It is possible that pigs that were playing were more likely to be involved in agonistic interactions due to the fact that they were awake and active. However, few studies have investigated play in pigs and those that have often focus on pigs younger than 4 weeks of age. Future studies exploring social play in pigs may provide insight about positive social relationships.

## 5. Conclusions

The first objective of this study was to quantify potentially affiliative behaviors in grow-finish pigs, to examine how these behaviors change over the weeks following a mixing event, and how these behaviors change in response to being moved to a new pen. The second objective was to assess the relationship between affiliative and agonistic behaviors. Non-agonistic physical contact, nosing, and play did change in the weeks following a mixing event, with pigs spending more time in contact with other pigs in their group at weeks 3 and 6 after mixing, more time nosing after mixing than at later time points, and playing most in week 3 after mixing. There appeared to be no effects of moving pigs to a novel pen on affiliative or agonistic behaviors. There were no predictive relationships between affiliative behaviors at mix and agonistic behaviors 3, 6, and 9 weeks later. However, non-agonistic contact at all time points was negatively related to aggression while nosing and play were associated with more aggression. The results of this study suggest that non-agonistic physical contact could be an indicator of positive social relationships and stability but the role of nosing and play in affiliation or social cohesion are less clear.

## Figures and Tables

**Figure 1 animals-12-00206-f001:**
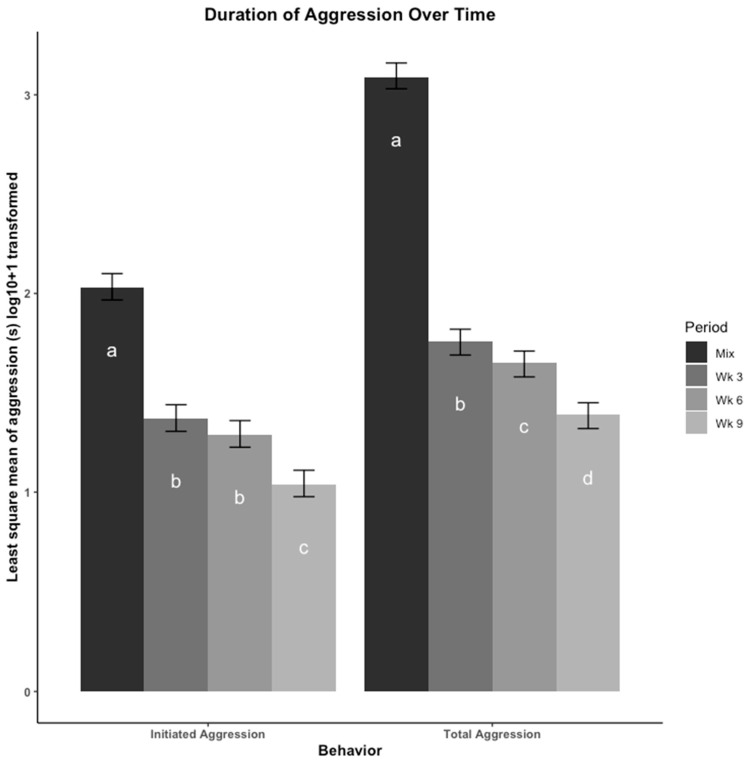
Aggressive behaviors were compared across four time points using least square means regression analysis. Aggressive behaviors were recorded using duration (seconds (s)) then log_10_ + 1 transformed. The four time points are: immediately after mix, and 3, 6, 9 weeks (wk) after mix. Observations at 6 wk were made after pigs were moved to a new pen with their group. Aggression is presented as total duration (s) of aggression and total duration (s) of initiated aggression. Errors bars represent the 95% confidence interval of the least square means. a–d: Bars with different letters denote time periods that showed significantly different durations of that behavior (*p* < 0.05, Tukey, HSD adjusted).

**Figure 2 animals-12-00206-f002:**
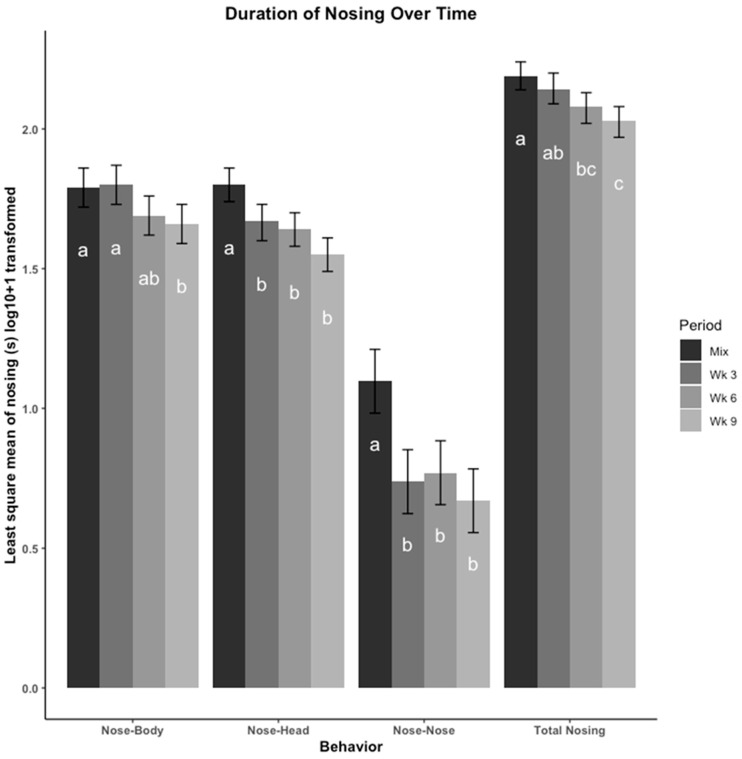
Nosing behaviors were compared across four time points using least square means regression analysis. Nosing behaviors were recorded in duration (seconds (s)) then log_10_ + 1 transformed. The four time points are: immediately after mix, and 3, 6, 9 weeks (wk) after mix. Observations at 6 wk were made after pigs were moved to a new pen with their group. Nosing is presented as total duration (s) of each type of nosing (nose-head, nose-body, nose-nose) depending on what part of the body the initiator pig’s nose was touching. Errors bars represent the 95% confidence interval of the least square means. a–c: Bars with different letters denote time periods that showed different frequencies of that behavior (*p* < 0.05, Tukey, HSD adjusted).

**Figure 3 animals-12-00206-f003:**
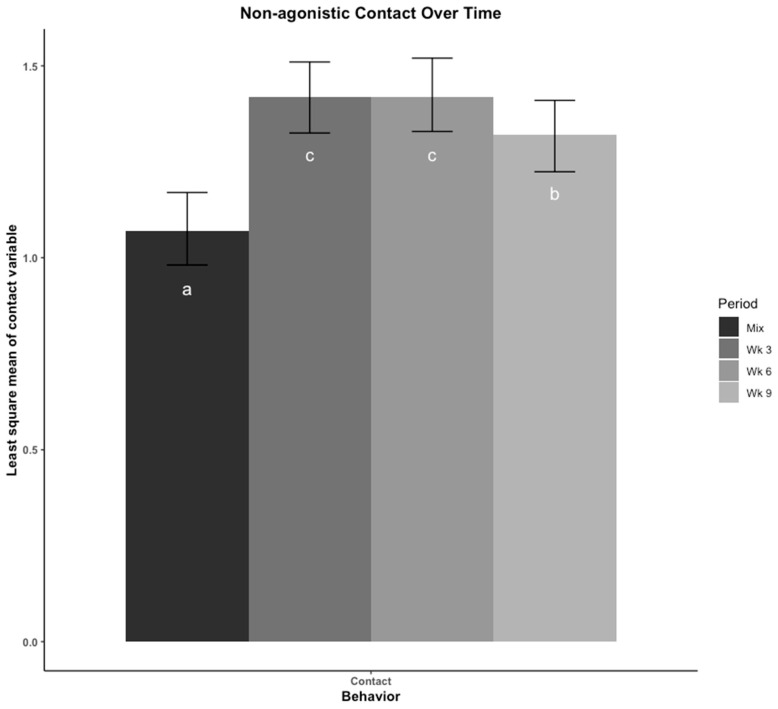
Non-agonistic contact was compared across four time points using least square means regression analysis. Pigs were observed for non-agonistic physical contact with conspecifics using scan-sampling every 10 min for 4 h. The contact variable was scored as the number of unique pigs the focal pig was in physical contact with without overt aggression occurring at each scan interval divided by the total number of scan intervals. The four time points are: immediately after mix, and 3, 6, 9 weeks (wk) after mix. Observations at 6 wk were made after pigs were moved to a new pen with their group. Errors bars represent the 95% confidence interval of the least square means. a–c: Bars with different letters denote time periods that showed different proportions of contact behavior (*p* < 0.05, Tukey, HSD adjusted).

**Figure 4 animals-12-00206-f004:**
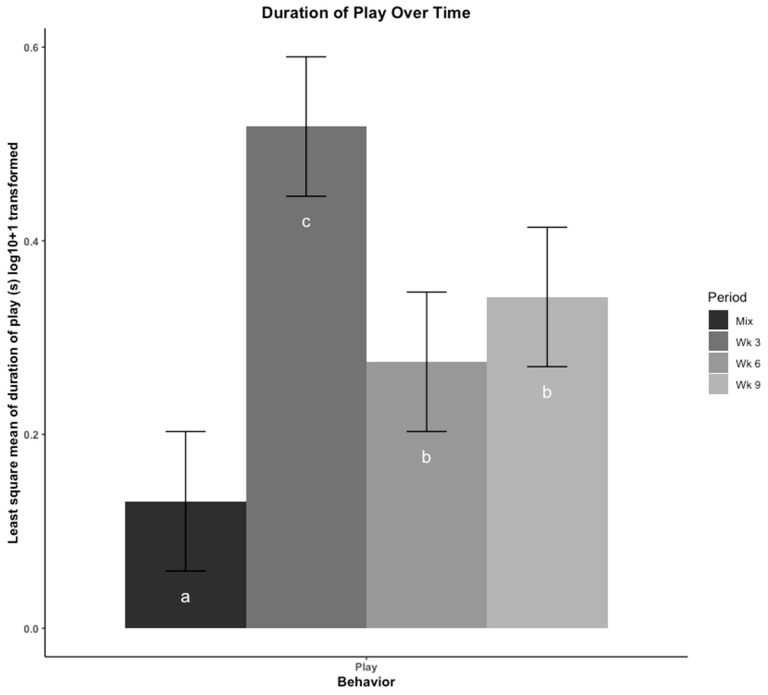
Play was compared across four time points using least square means regression analysis. Play was recorded in duration (seconds (s)) and log_10_ + 1 transformed. The four time points are: immediately after mix, and 3, 6, 9 weeks (wk) after mix. Observations at 6 wk were made after pigs were moved to a new pen with their group. Play is presented as total duration (s). Errors bars represent the 95% confidence interval of the least square means. a–c: Bars with different letters denote time periods that showed different frequencies of play behavior (*p* < 0.05, Tukey, HSD adjusted).

**Table 1 animals-12-00206-t001:** Ethogram of aggressive behaviors.

Behavior	Definition
Damaging aggression	Periods of interaction during which bites are delivered at an approximate rate of greater than or equal to 1 bite per 3 s.
Reciprocal fight	Both pigs are engaged in damaging aggression for greater than or equal to one second. Pigs may be pressing against each other and head knocking during this time.
Rest during fight	Damaging aggression was given/received whilst the recipient was resting for greater than or equal to 3 s during a reciprocal fight (reciprocal fighting must occur before and after this event for it to be classified as a rest during fight). The recipient does not show damaging aggression during this period.
Withdrawal	One pig tries to leave a reciprocal fight but the other pig continues to give damaging aggression at a rate of greater than 1 bite per 3 s. The recipient does give any damaging aggression for greater than 3 s. Then, there must be a period of no interaction between the pigs for at least 3 s after this.
Attack	One-sided damaging aggression was given by one pig without the recipient returning damaging aggression during the attack. The bite rate should be greater than 1 bite per 3 s. Recipient pigs cannot deliver damaging aggression back to the attacker for the 3 s before or after the attack.
Single bite	A pig delivers a knock with the head or snout against the head, neck, or body of the other pig with the mouth open. A single bite should only be recorded when it occurs at least 5 s before or after a period of damaging aggression.
Head knock	A rapid, forceful thrust upward or sideways with the head or snout against any part of the body of recipient pig with the mouth shut.
Inverse pressing	Two pigs stand side by side facing front to back. Pigs push their shoulders hard against each other, throwing the head against the neck and flanks of the other. Pigs may be biting each other at a rate of less than 1 bite every 3 s.
Parallel pressing	Two pigs stand side by side with heads in the same direction. Pigs push hard with the shoulders against each other, throwing the head against the neck or head of the other. Pigs may be biting each other at a rate of less than 1 bite every 3 s.

**Table 2 animals-12-00206-t002:** Behaviors of non-agonistic contact (proportion of time in non-aggressive contact with unique pigs at each time point), duration of nosing (seconds (s)) (scaled to adjust for differences in measurement between behaviors), and duration of play (s) were compared to total duration of aggression (s) and total duration of initiated aggression (s) at four time points: immediately after mixing, and 3, 6, and 9 weeks (wk) after mixing. Comparisons were made using linear mixed models.

			Estimate	SE	F_(1, 4)_	*p*
Total aggression (s)	Mix	Contact	−0.272	0.072	13.938	<0.001
		Nosing	−0.139	0.028	24.029	<0.001
		Play	−0.020	0.005	18.743	<0.001
	3 wk	Contact	−0.118	0.044	7.001	0.009
		Nosing	0.075	0.019	14.145	<0.001
		Play	0.004	0.002	4.619	0.033
	6 wk	Contact	−0.165	0.039	17.264	<0.001
		Nosing	0.066	0.018	13.086	<0.001
		Play	0.004	0.002	3.941	0.048
	9 wk	Contact	−0.179	0.054	11.081	0.001
		Nosing	0.087	0.022	15.521	<0.001
		Play	0.003	0.003	1.854	0.175
Total initiated aggression (s)	Mix	Contact	−0.254	0.078	10.276	0.001
		Nosing	−0.099	0.029	11.042	0.001
		Play	−0.018	0.005	12.422	0.001
	3 wk	Contact	−0.176	0.053	10.764	0.001
		Nosing	0.073	0.024	9.268	0.003
		Play	0.005	0.002	5.612	0.019
	6 wk	Contact	−0.159	0.048	10.562	0.001
		Nosing	0.066	0.023	8.602	0.004
		Play	0.002	0.003	0.534	0.466
	9 wk	Contact	−0.169	0.058	8.351	0.004
		Nosing	0.086	0.024	12.546	<0.001
		Play	−0.002	0.003	0.586	0.445

## Data Availability

The data presented in this study are available on request from the corresponding author.

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
