# Peer review of "The Social Life of Pigs: Changes in Affiliative and Agonistic Behaviors following Mixing"

_animals, 2022, doi:10.3390/ani12020206_

Round 1

Reviewer 1 Report

The social life of pigs: can understanding of affiliative behavior help reduce aggression in groups?

The manuscript by O'Malley et .al.  performed a systematic analysis at different time point of mixing the group-housed pigs to investigate their aggression/agonistic behavior. In my opinion, the manuscript is quite interesting and important which clearly highlight the importance of systematic monitoring of animals to understand their complex interplay as well as to improve the animal welfare.  In total, I find the design of the methodology is very valuable, especially it is written in a very clear way.  In particular, the usage of generalized linear mixed models is very crucial in their design. Although, I highly appreciate the work of the authors, there are still some major & minor issues with the manuscript with its current state that are described in detail below.

First, I am wondering why the authors completely ignored novel development using machine learning based methods to automatically register the behavioral observations. At least a short discussion is needed for computer vision based monitoring of animals

Please see these studies:

Study 1: https://doi.org/10.3390/s20133670

Study 2: https://doi.org/10.3390/s19040852

Study 3: https://doi.org/10.3390/s21227512

My another major concern is the publishing of the data set also video materials. It must be free available for the scientific community otherwise, no-one could ever re-construct this study/results again. Especially, this type of the analysis and data sets could be crucial for the improving of the animal welfare in further investigations, but in current study design without video materials no one can use it directly or study different aspect of the animal behavior. Consequently, the authors should please provide the dataset publicly.  I am also aware about the data-safety restrictions, but the authors can anonymize some information in the data, if it is necessary.

Minor concern: 

in line 75: Please change the order of the references:  from [12,10,13,14,7] to [7,10,12,13,14]

in line 81: The objectives of this research were to … -> The objectives of this research is to

Reviewer 2 Report

O’Malley et al. The social life of pigs: can understanding of affiliative behaviour help reduce aggression in groups?

Reviewer comments

General

This research examined changes in affiliative behaviour between growing-finishing pigs over days post-mixing. It is of interest because most research focuses on changes in aggressive behaviour due to its negative effects on animal welfare, but focusing on affiliative behaviour provides opportunity to promote positive welfare states.

I have a few general comments, with specific comments also included below. Firstly, I think the title of the paper and the introduction are a bit misleading, because this research examined changes in affiliative behaviour and did not really examine how affiliative behaviour could be used to reduce aggression nor is any advice provided in how to promote affiliative behaviour in these systems. Second, the authors state in the methodology that some animals were familiar in the pen. This is a very interesting part of the study, particularly in terms of timeframe to achieve social cohesion and engagement in positive social behaviours. The authors could have examined all the variables they recorded, but in terms of “familiar” or “un-familiar” animals. This would be more useful, and would reveal if changes in affiliative behaviour over time were due to increased amounts delivered to “unfamiliar” pigs, suggesting cohesion and social stability. Finally, while significant, the estimates for the strength of the relationships between play and aggression (and even nosing and aggression) were very low, and even negligible. The authors do not acknowledge this, and I think the low strength of the relationships make parts of their discussion and conclusion redundant.

Simple summary

L22 “affiliative behaviours changed” – in what way did they change?

L27-28 Don’t your results show that familiarity promotes social contact and other positive social behaviours? I.e., agonistic behaviour highest, social contact lowest and play lowest at mixing, when pigs were largely unfamiliar. Then as pigs became familiar with one another, agonistic behaviour declines and social contact/play increase.

Abstract

L32 At what age were pigs mixed? What proportion of familiarity (if any)?

L34 What do you mean by “social contact”? Is this the non-agonistic physical contact, or does it also include nosing etc.

Introduction

General

My general comment about the introduction is that I think it discusses aggressive/affiliative behaviours in a way that makes them seem like a cause of social unrest/cohesion rather than a symptom or effect of social unrest/cohesion. For example, L54-56 (“affiliative behaviours occurring among pigs that could be supposed to improve group cohesion and stability”), and also L62 (“use it to form stable social groups”). I think the group cohesion and stability come first, and increase affiliative behaviour will follow. So really anything that promotes fast establishment and successful operation of social dominance relationships is key – i.e., proportion of familiar animals, access to high priority resources, socially competent animals etc. (I’m sure you already know all of this). I wonder if your introduction would better position your findings by discussing affiliative behaviour in terms of promoting positive welfare states, rather than simply minimising negative states associated with aggression, with a secondary aim to understand change in affiliative behaviour as a measure of social integration or group cohesion.

Specific comments

L52 “rather delay its onset” Sometimes, but sometimes they can reduce the severity and duration of high intensity aggression post-mixing. It depends on the strategy being used (e.g., mixing at night or olfactory suppressants versus increased floor space per pig). I’m not sure we will ever eliminate aggression while we are mixing unfamiliar pigs – some aggression is a normal behavioural response and needed to establish new dominance relationships. But unless we give the pigs enough time, space, and other resources (e.g., barriers) very high and prolonged levels of aggression will be observed.

L58 In this study how long were pigs observed post-mixing?

L61-63 I don’t think you have made the case for a genetic basis to affiliative behaviour.

L65-71 I don’t see the point to this paragraph – you have discussed aggression in the first paragraph, and this doesn’t seem to add anything new

L76 You have missed a ( before the word “However”

L76-77 “behaviours are truly affiliative” and “positive social interactions”. I think this is a good point, particularly with reference to nosing. Could you use nosing as a clear example here?

Materials and methods

L104-112 Here you mention there is some familiarity within groups (about 1/3). This is an essential part of your work as, I believe, familiarity is one of the key components to achieving social stability in pigs. This needs to be in your abstract also. Could affiliative behaviour delivered to familiar versus unfamiliar animals over time be an indicator of social integration?

L120-121 Please confirm that you observed the pigs for 4 h immediately post-mixing and immediately after being moved to the new pen in week 6. Was the new pen comparable in dimensions and design?

LL123 From my experience, a lot of the play behaviour observed after mixing young pigs has agonistic components, especially head knocks. Sometimes these escalate from play into a fight. How did you deal with these? I understand you didn’t include social play for this reason, but what about aggression? Did you record head knocks given during a bout of locomotor play (leaps, scamper, pivot etc) as aggressive?

L124 Did you record frequency of play bouts also? Or just duration?

L128-130 Did you have a bout criteria for nosing behaviour? E.g., If a pig made a brief contact, was it still included as nosing? Did you record whether it was delivered to familiar or unfamiliar penmates?

L132 How do you define “mutually” touching? Do you mean that all nose-to-nose were assumed to be mutual, because of the difficulties identifying an initiator?

L136-139 Does this mean any part of the body in contact? And were these mutually exclusive from nose-nose? For example, if there was a nose-nose contact, was this recorded as a nose bout as well as a physical contact? Was it always easy to see when there was physical contact from your videos?

L141 Can you have a non-reciprocal fight? By its nature, a fight is reciprocal, isn’t it? Do you mean reciprocal aggression was defined as a fight? Also, what is the definition of an attack? Head knocks – does this include those delivered during play bouts? It would also be good if you could be a bit more explicit in the duration of total aggression. Was this all aggression delivered + all aggression received by an individual pig? Whereas initiated was all bouts of aggression delivered (excluding fights) by the individual pig? What if a pig reciprocated only once? Is that counted in either of these measures, or was it counted as a fight? Often I have seen a few single bouts escalate into a mutually delivered/received aggressive interaction (fight) – how did you handle that?

L147-148 Did you analysis account for repeated observations of the same pigs/pens over time?

Results

Figure 1. y-axis – can you please be clear that these values are “seconds per pig”. The title of the figure needs to be more explicit also “Duration of aggressive interactions” or something like that. As an aside, 3s total aggression is not very much for newly mixed pigs – am I interpreting this right? The variables you have reported, initiated and total aggression. I have been struggling to understand why you would expect these two variables to be different? Perhaps I am missing something because I wasn’t clear on their definitions in your methods. But the initiated aggression per pig in a pen, would surely be comparable to the total aggression per pig in a pen? As the are delivering aggression to someone. This is indeed what your figure shows, that they are reporting the same changes over time.

Figure 3. In the caption you call it social contact – but I think you means “physical” contact. X-axis; behaviour “contact” – can you please be more descriptive? What kind of contact? And units – seconds, frequency, proportion of observations. Similarly, the title could be more explicit “Non-agonistic physical contact over time), for example. I think you could get more from this data from looking at who they were in contact with over time – familiar or not.

L230-232 Some of these estimates are very low – especially those related to play. You need to make this clear in your text. While statistically significant, I am doubtful of the impact of these relationships.

L233 What is “affiliative social contact”? Did you report on this earlier, I don’t recall its definition?

L240-244 I struggle with this, mostly because I think it sounds again like the affiliative behaviour is predictive of social cohesion rather than social cohesion being predictive of affiliative behaviour. Perhaps if you looked at affiliative behaviour delivered to unfamiliar pigs and aggression that would be more revealing, as I think this is different story to affiliative behaviour towards familiar conspecifics at mixing.

Table 1. “Affiliative behaviour of social contact (proportion of time in non-aggressive contact” – I think this is a bit presumptuous. Time in non-aggressive contact may or may not be affiliative, it could be exploratory, or simply having a foot touch while lying. I’m also not sure what “scaled to adjust for differences in measurement between behaviours” means? Here again, total aggression and total initiated are telling the same story – why do you need both?

Discussion

L255-263 I think this discussion on aggression is a bit negative. Overall, I would think that aggression is useful in understanding potential adverse effects on welfare, due to the effects of aggression on injury and pain, for example. However, this doesn’t mean that the absence of aggression is indicative of good welfare, and the reference you provide is supportive of this. On the flip side, the understanding of socially affiliative behaviour has the potential to promote positive states of welfare. So, I see both as being important, reduction of negative effects caused by aggression AND promotion of positive states such as those brought by affiliative behaviour. However, this discussion makes it sound like it’s an either/or scenario.

L275 This point about proximity to familiar or preferred conspecifics; this is why I think your measure of social contact is one of social cohesion/integration, and why you need to report on familiarity or not. I see you have made this point at L277-279 – I think this is likely rather than possible. But you could assess this by splitting social contact by familiarity

L279-282 Why isn’t this data reported here? As you are only referring to a conference abstract the data could be incorporated into this paper to tell a more complete picture. If not, I think you need to bring in this reference earlier as I had many questions about familiarity as I was reading your paper. You probably could make it really clear in your introduction that you are building on your previous work if you are not incorporating the data into this paper.

L289 I’m still not clear on the benefits of separating total and initiated aggressive interactions, indeed, they told pretty much the same story in your results.

L290 One limitation to looking at proximity is that distance between animals can be limited by the physical dimensions of the pen – you could mention this.

L328-330 I think this is an important methodological detail – can you pleases state this when describing your behavioural observations in Section 2.2

L330-331 I think you need to be cautious with the impact of this relationship, as the estimates were very low (0.004-0.02), so the contribution that variation in play to variation in aggression is negligible. In light of these weak relationships, I think most of this discussion on relationships between aggression and play is redundant.

L355-356 Again, these relationships were very weak

Reviewer 3 Report

This is an interesting manuscript by O’Malley et al that examines the often over looked role of positive interactions between pigs in their social dynamics.  A straightforward observational behavior experimental design is applied across pens of growing pigs at different stages of their life cycle and then interactions between both positive and negative behaviors are examined.    As expected on a population level agonist behaviors decreased over time in static pens and some, but not all, affiliative behaviors increased.   When comparing agnostic and affiliative behaviors at the individual animals certain positive behaviors were associated with reduced agnostic behaviors while others yielded more agnostic behaviors.  Finally, duration of these affiliative behaviors were not predictive of agnostic outcomes.  Taken together these findings highlight the gaps in our knowledge about what has been described as affiliative behaviors in pigs.

The main concern with this work is semantics.  The term “group-housed pigs” is at best redundant and at worst duplicit and needs to be expunged from the manuscript.  This reviewer knows of no time when growing pigs like those that are used in this study (age 10 to 19 weeks of age) are housed individually and so it is hard to understand the relevance of this qualifier as such should be removed. It is much like the use of the advertising term “gluten-free milk” as all milk is gluten and thus there is no new information in the gluten-free descriptor and has potential ascribe false attributes to those that are less familiar.

 My concern is that this term, “group-housed pigs” creates confusion in the literature and should not be encouraged. There are times in pork production where animals, particular sows, commonly are housed in either groups or individual stalls and the distinction is important. This is not the case with growing pigs and thus I find no justification for its use. The fact that sows are so commonly housed as individuals, but grower pigs are not, likely speaks to differences in the social dynamics within populations of breeding animals and growing pigs that should not be muddled.   If the authors want to assert the notion that these studies on growing pigs provide possible insight into our understanding of the social dynamics of older pigs (sows) then they should do so by developing these ideas in the discussion and not by spuriously mislabeling their study animals.  I understand that there is already precedent in the literature for this confusing and misguided nomenclature but it does not make it right and I needs to be corrected here. 

Round 2

Reviewer 1 Report

The manuscript by O'Malley et .al. was improved based on comments, but the authors refused to address my major concerns. Instead, I was provided with several studies that I need to see to understand exactly what the authors mean by this.  Normally, I might assume that the authors would add their opinion on machine learning to the discussion. However, they have declined to do so.

The second important comment is about the dataset (videos) that must be available to the scientific community. Otherwise, we have to directly believe the authors' observation without seeing any proof of concept. I think this point is inevitable to evaluate the results and discussion in this manuscript. This option is not currently offered by the authors, so the results are not currently provable. At least the authors could provide a small part of their data (e.g. ~250 GB) as these guys did: please see the link: https://drive.google.com/drive/folders/1C_wABDzfpdaRykVHoWSN8vAaLXs8Yaxn

Instead, the authors propose to make data available in the future. But, as I mentioned before, without seeing the data, it is not acceptable for me to accept the results only based on the authors' statement.

Author Response

My co-authors and I are concerned that only 1 of 3 initial reviewers has examined our revisions, which were considered major.   The reviewer who has responded is fixed on two points and has again requested major revisions.   I would appreciate your perspective on these as my co-authors and I feel that the requested revisions are not directly related to the quality of the manuscript or to the research we conducted and the data we present.   Point 1 = making our data sets and video publicly available. We can certainly make the spreadsheet of our data available. We decoded >50TB of video for this manuscript, which is more than most data sharing repositories can handle. The reviewer has indicated that sharing a portion of the video would be sufficient, yet from a scientific perspective, I and my co-authors are not sure how exactly the reviewer or others would then be able to check the accuracy of our analysis using only a small portion of the video. Does Animals require that we share all of our video data? If so, does Animals have a repository we can use to house >50TB of video data? If not, can this request from the reviewer be ignored?   Point 2 = requiring a discussion of the use of machine vision to decode video data. My co-authors and I are experts in precision livestock farming and regularly do research to develop machine vision tools, and we are well aware that the machine vision tools don't exist to decode social behavior of individual pigs even under the research farm conditions we used. Most computer vision studies are focused on development of methodology and only use a handful of animals, because at this point, that is as much as the technology can handle. Thus, we manually decoded our video. Our paper has nothing to do with video decoding methodology but rather is about analyzing the social lives of pigs. We are not sure where or how such a discussion of machine vision technology as a method of decoding video fits into our paper or what relevance it has to our findings on the social behavior of pigs. Discussing the feasibility of automating detection of behavior in nearly 1000 pigs using computer/machine vision would be an entirely different paper and involve an entirely different data analysis.   Neither of the points raised by the only reviewer who responded in this round were raised by the other 2 reviewers. Thus, this single individual (1/3 of the reviewers) is now dictating all points we must address in our revisions.   We would appreciate either a decision from you as editor or from an additional reviewer on these two points. Ideally, we'd like to see the other initial reviewers reinvited again to see if we addressed their salient recommendations. Now that the holidays have passed, they may be more able or willing to complete a review of our revisions. If not, perhaps you as editor could also assess whether we addressed the majority of the revisions accordingly.

Reviewer 2 Report

I thank the authors for their consideration of my comments, and now recommend this manuscript for publication.